# Electromechanically reconfigurable optical nano-kirigami

Shanshan Chen [1,4], Zhiguang Liu[2,4], Huifeng Du [3,4], Chengchun Tang [2,4], Chang-Yin Ji[1], Baogang Quan[2], Ruhao Pan[2], Lechen Yang[2], Xinhao Li[3], Changzhi Gu [2], Xiangdong Zhang[1], Yugui Yao[1], Junjie Li [2✉], Nicholas X. Fang[3✉] & Jiafang Li [1,2✉]

Kirigami, with facile and automated fashion of three-dimensional (3D) transformations, offers an unconventional approach for realizing cutting-edge optical nano-electromechanical systems. Here, we demonstrate an on-chip and electromechanically reconfigurable nano-kirigami with optical functionalities. The nano-electromechanical system is built on an Au/ SiO₂/Si substrate and operated via attractive electrostatic forces between the top gold nanostructure and bottom silicon substrate. Large-range nano-kirigami like 3D deformations are clearly observed and reversibly engineered, with scalable pitch size down to 0.975 μm. Broadband nonresonant and narrowband resonant optical reconfigurations are achieved at visible and near-infrared wavelengths, respectively, with a high modulation contrast up to 494%. On-chip modulation of optical helicity is further demonstrated in submicron nano-kirigami at near-infrared wavelengths. Such small-size and high-contrast reconfigurable optical nano-kirigami provides advanced methodologies and platforms for versatile on-chip manipulation of light at nanoscale.

[1] Key Lab of Advanced Optoelectronic Quantum Architecture and Measurement (Ministry of Education), Beijing Key Lab of Nanophotonics & Ultrafine Optoelectronic Systems, and School of Physics, Beijing Institute of Technology, Beijing, China. [2] Institute of Physics, Chinese Academy of Sciences, Beijing, China. [3] Department of Mechanical Engineering, Massachusetts Institute of Technology, Cambridge, MA, USA. [4]These authors contributed equally: Shanshan Chen, Zhiguang Liu, Huifeng Du, Chengchun Tang. ✉email: jjli@aphy.iphy.ac.cn; nicfang@mit.edu; jiafangli@bit.edu.cn

The state-of-the-art practice of cutting, folding, bending, and twisting flat objects into versatile shapes, named kirigami or origami (origami does not include the cutting process)[1,2], has recently arisen as a facile and automated fashion of three-dimensional (3D) manufacturing[3–7]. The fascinating transformation of two-dimensional (2D) precursors into complex 3D architectures has enabled exceptional geometries and functionalities[8,9], which arouses great interests in the areas of microelectromechanical systems (MEMS)[10–13], extraordinary mechanics[14–16], biomedical devices[17], acoustic materials[18], energy storage systems[19,20], microwave metamaterials[21,22], and terahertz spectroscopy[23]. Particularly in the microscale/nanoscale region, kirigami/origami has achieved artful 3D nanomanufacturing[6,24] without the need of spatial translation[25,26] or multilayer stacking[27] in traditional on-chip 3D microfabrications. More importantly, compared with its mesoscopic counterpart[5], the nanoscale kirigami (named nano-kirigami) is highly desirable for the excitation of optical resonances, which opens an avenue for optical kirigami/origami. For example, nano-kirigami and related techniques have been employed to generate functional photonic nanostructures, such as elastic wide-angle gratings[28], Fano-resonant metamaterials[29], diffractive meta-surfaces[30], toroidal metamaterials[31], reversible mid-infrared switchings[32], chiral optical materials[24], etc. However, the conventional nano-kirigami methods are mainly based on suspended lift-off films[6,24], long-span film windows[31], or elastic substrates[28,33], of which the platforms face challenges[34] in large-scale, uniform, and integrable 3D nanomanufacturing that are valued in real-world applications.

Another important characteristic of kirigami/origami is its reconfiguration capability based on the reversible displacement of the transformable component[21,22,35]. This type of features has been widely employed in optical MEMS[36,37], like the movable micromirrors of digital micromirror devices (DMD)[36] commercialized successfully in digital light processing and related 3D printing industry. Since the speed of a mechanical actuation is fundamentally limited by $\omega = \sqrt{k_{eff}/m_{eff}}$ ($k_{eff}$ and $m_{eff}$ are effective stiffness and mass of the equivalent mass–spring system, respectively)[38], it is very desirable to scale down the reconfigurable mechanical component and increase the effective stiffness through nano-kirigami (Supplementary Fig. 1). Furthermore, in the nanoscale region, the electromagnetic and displacement fields are highly confined and the optical, electrical, and mechanical interaction can be dramatically enhanced, which could form advanced nano-opto-electromechanical systems (NOEMS)[38] that are promising for photonic circuits, optical switches, quantum devices, etc. However, in electromechanical systems, there is always a trade-off between the miniaturization of reconfigurable unit and the enhancement of modulation depth (determined by the spatial displacement), which limits the designs of NOEMS to only a few options (like the ultrathin cantilevers and membranes)[38] and makes it highly challenging for submicron pixelated manipulation.

Here, we demonstrate an on-chip and electromechanically reconfigurable nano-kirigami with optical functionalities. The nano-electromechanical system is built on an Au/SiO$_2$/Si chip, where the electrostatic forces between the top suspended gold patterns and bottom silicon substrate actuate the 3D nanokirigami transformations. With flexible nano-kirigami designs, broadband nonresonant and narrowband resonant optical reconfigurations are demonstrated at visible and near-infrared wavelengths, respectively. By scaling down the nano-kirigami units to submicron sizes, resonant optical reconfiguration with high contrast up to 494% is achieved. On-chip modulation of optical helicity is also realized in submicron nano-kirigami at near-infrared wavelengths. Such small-size and high-contrast reconfigurable optical nano-kirigami could provide very useful

methodologies and platforms for interesting physics and advanced applications in nanophotonics, optomechanics, MEMS, NOEMS, etc.

## Results

**Scheme for reconfigurable nano-kirigami.** In-plane and out-of-plane displacement are two schemes that are normally adopted in electromechanical photonic devices. The former one, such as the use of electrostatic comb drives[39] and parallel strings[40], can produce a large range of deformation, but lacks locally pixelated manipulation. The latter one, by using thermal[41], electric[35], magnetic[42], or nonlinear effects[43], has been demonstrated in a few metamaterials[44], which were subject to further improvements in modulation depth, unit miniaturization, and displacement range toward practical applications. Here, an electromechanical nano-kirigami is proposed to achieve pixelated out-of-plane deformations with large range and modulation depth. For a specific illustration, the reconfiguration scheme starts from an array of 2D gold pinwheels suspended above SiO$_2$ pillars, as schematically shown in Fig. 1a. When a proper voltage is applied, electrostatic force will be introduced between the top suspended nanostructures and the bottom silicon substrate (Fig. 1a–c). Similar to the general case of an electromechanical capacitor with conductive plates, the initial electrostatic force can be written as $F_e = \frac{1}{2}V^2 \frac{\partial C}{\partial d} = -\frac{1}{2}V^2 \frac{\varepsilon A}{d^2}$ (ref. 38), where $C$ is the capacity under $C = \frac{\varepsilon A}{d}$, $V$ is the applied voltage, $A$ is the effective area of the plates, and $d$ and $\varepsilon$ is the thickness and permittivity of the materials in between the plates, respectively. As the conductive plates are replaced by 2D nano-kirigami patterns in this work, the distribution of the local force (represented by the stress $\sigma = F_e/A$) is varied conforming to the topography and boundary of the 2D nanopatterns. When the stress $\sigma$ and induced torque are strong enough, the 2D patterns (Fig. 1a) will be deformed into 3D geometries (Fig. 1b) based on nano-kirigami principles[24], like the illustrations in Fig. 1c, d and scanning electron microscope (SEM) images in Fig. 1e, f. In such a way and under the restoring mechanical forces ($F_r = k_{eff}\Delta d$), reconfigurable nano-kirigami transformations can be readily achieved by switching on and off the voltage.

**Realization of electromechanical nano-kirigami.** To realize the electromechanical nano-kirigami, a 2D nanopatterning process is used to represent the "cutting" step and the electrostatic force is employed to trigger the 3D "folding" process subsequently. Specifically, a commercial SiO$_2$/Si substrate coated by a 60-nm-thick gold film is processed with the standard electron-beam lithography (EBL) and wet-etching process, as schematically shown in Fig. 2a (see "Methods"), which is compatible with the complementary metal-oxide–semiconductor technique when the Au layer is replaced by proper conductive nanofilms (such as Si). The EBL exposure can result in nano-kirigami patterns as large as $500 \times 500 \ \mu m^2$ (Fig. 2b) and is replaceable by commercial ultraviolet exposure for massive production. It should be mentioned that the following wet-etching process, aiming to locally suspend the 2D patterns that are deformable upon later electrostatic forces, is critical. First, the wet-etching speed should be accurately controlled. Over-etching can result in the lift-off or collapse of the top gold patterns (Supplementary Fig. 2). A successful etching condition is illustrated in Fig. 2c, where the bottom SiO$_2$ supporters are clearly observed after the removal of top structures. Second, due to the small thickness of the SiO$_2$ layer (500, 300, or 200 nm in this work), capillary pressure arises due to the surface tension of the droplet upon drying. This may cause the suspended structure to stick onto the silicon substrate if its effective stiffness is small, such as the type-iii spiral structure shown in Fig. 2d, e.

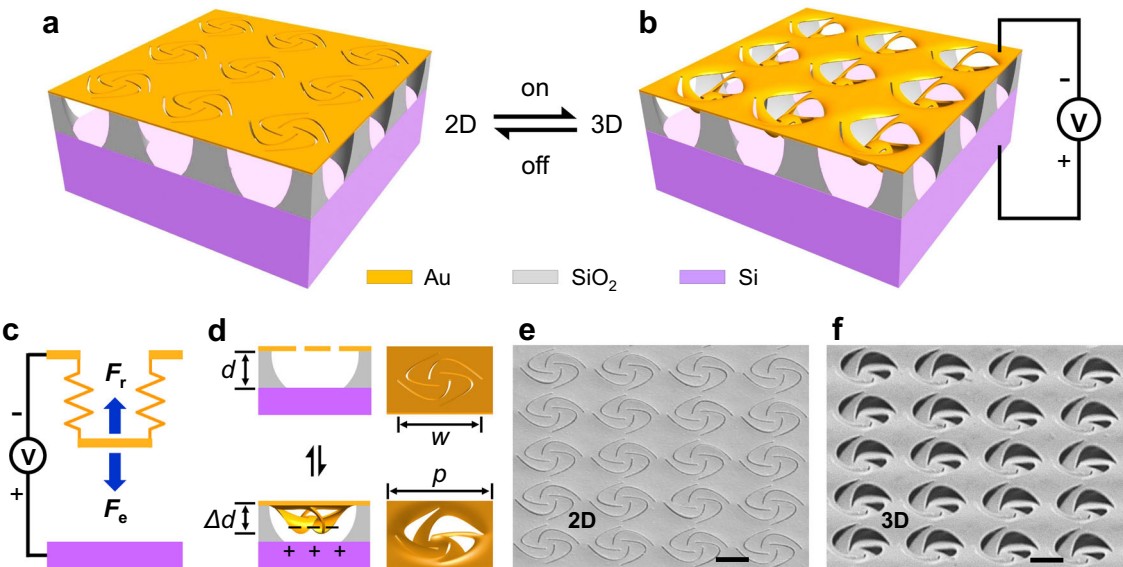

**Fig. 1 Scheme for reconfigurable nano-kirigami. a, b** Schematic of **a** a 2D pinwheel array and **b** its downward 3D state under attractive electrostatic forces when the voltage is on. Each gold pinwheel is locally suspended by four SiO$_2$ supporters with thickness of $d$. **c** A simplified electromechanical model of the reconfigurable nano-kirigami, in which the displacement of the suspended nanostructure is controlled by the downward electrostatic force ($F_e$) and upward mechanical restoring force ($F_r$). **d** Front-view and side-view plots of (top) 2D and (bottom) calculated 3D deformed pinwheel in the gold layer. **e, f** Side-view SEM images of as-fabricated 2D pinwheels and the downward deformed 3D pinwheels after applying DC voltage of $V = 65$ V. Structural parameters: gold thickness: $t = 60$ nm; pinwheel width: $w = 2$ μm; lattice periodicity: $p = 2.25$ μm; $d = 500$ nm. Scale bars: 1 μm.

Therefore, geometric optimizations of the nano-kirigami designs (Supplementary Figs. 1–4) are necessary to avoid the negative effects of the capillary forces, such as the succeeded types-i/-ii spirals in Fig. 2d, e. Finally, before the electrical reconfiguration, the deformable characteristics of the suspended 2D patterns are tested by the low-dose focused ion beam (FIB) irradiation-induced tensile stress[24], which will induce upward deformations if the fabrication is successful, as the types-i/-ii spirals shown in Fig. 2e and the pinwheels shown in Fig. 2f.

As long as the deformable 2D nanopatterns are successfully suspended onto SiO$_2$ supporters, the chips are bonded on electric boards (Supplementary Fig. 5a, b), on which the freely suspended nanostructures can be pulled downward by attractive electrostatic force when a proper voltage is applied, as illustrated in Fig. 1a–d. The deformations in turn perturb the electrostatic field and lead to the redistribution of stress $\sigma$ within the nanostructures until a new equilibrium state is achieved, where the structural stiffness and electrostatic force are balanced. Therefore, under low voltage where the effective $\sigma$ doesn't exceed the yield strength ($Y_g$) of the nanostructure, elastic deformations dominate and can be utilized for reversible reconfiguration (see Supplementary Movie 1). By contrast, under high voltage where the effective $\sigma$ largely exceeds $Y_g$, irreversible plastic deformations occur at the pull-in state. In this case, the induced permanent changes provide visual confirmation of the downward structural deformations, as shown by the difference between Fig. 1e, f and the uniform 2D-to-3D transformations in Fig. 2g (also see Supplementary Fig. 5f). It should be mentioned that these permanently downward deformed pinwheels at the pull-in state can be deformed upward by the low-dose FIB irradiation (Supplementary Fig. 5g and Supplementary Movie 2), in contrast to the capillary force-induced strong sticking (Fig. 2e and Supplementary Fig. 3c).

**Optical reconfigurations.** On-chip reconfigurable manipulation of light at nanoscale is one of the most important challenges faced by urgent applications, such as photonic integration, meta-surfaces, and optical metamaterials. Nano-kirigami can result in dramatic vertical displacements through actuating the

transformable unit, thus offering an inspirational methodology for reconfigurable photonics. Here, the nano-electromechanical deformations can induce two types of changes in optical responses. In the first case, where the working wavelengths are much smaller than the structural units, each pixel of the structure behaves like a deformable mirror to deflect the light in broadband. To test this scheme, a pinwheel array with periodicity of 2.5 μm is simulated under normal incidence at visible wavelengths. As plotted in Fig. 3a, the reflection spectrum drops significantly when the 2D pinwheels are deformed into 3D with a height of 300 nm under a voltage of 31 V, which is mainly caused by the diffraction to other directions, as shown by the inset of Fig. 3a and Supplementary Fig. 6a. The experimental reflection spectra plotted in Fig. 3b, indeed, exhibit dramatic changes with the increase of direct current (DC) voltage ($V$), due to the increased deformations. Such strong modifications in reflection occur in broadband from 400 to 1100 nm, and the maximum amplitude of modulation contrast reaches 51% at wavelength 750 nm (Fig. 3c).

Interestingly, it is observed that the reflection spectrum stops decreasing when $V > 32$ V (Fig. 3b, c), indicating that the maximum vertical deformation $\Delta d = d = 300$ nm is reached, which is very close to the simulation condition ($V = 31$ V) in Fig. 3a. Meanwhile, electromechanical simulation results show that the pull-in voltage of such nano-kirigami unit is about $V_p = 35$ V (Fig. 3d), below which the elastic deformation is induced. This is verified by the observation that the modified spectra turn back to the initial position after turning off the voltage at 35 V, as plotted in Fig. 3b, which provides an effective scheme for reversible optical modulation. For example, the modulation in optical reflection can be repeatedly switched by turning on and off the voltage, as plotted in the inset of Fig. 3c (also in Supplementary Fig. 7c–e). It is worthwhile to note that the modulation contrast is found highly dependent on the topographies of the 2D patterns (Supplementary Fig. 7), which, in turn, provides a useful approach for tailoring the electromechanical properties by geometric designs.

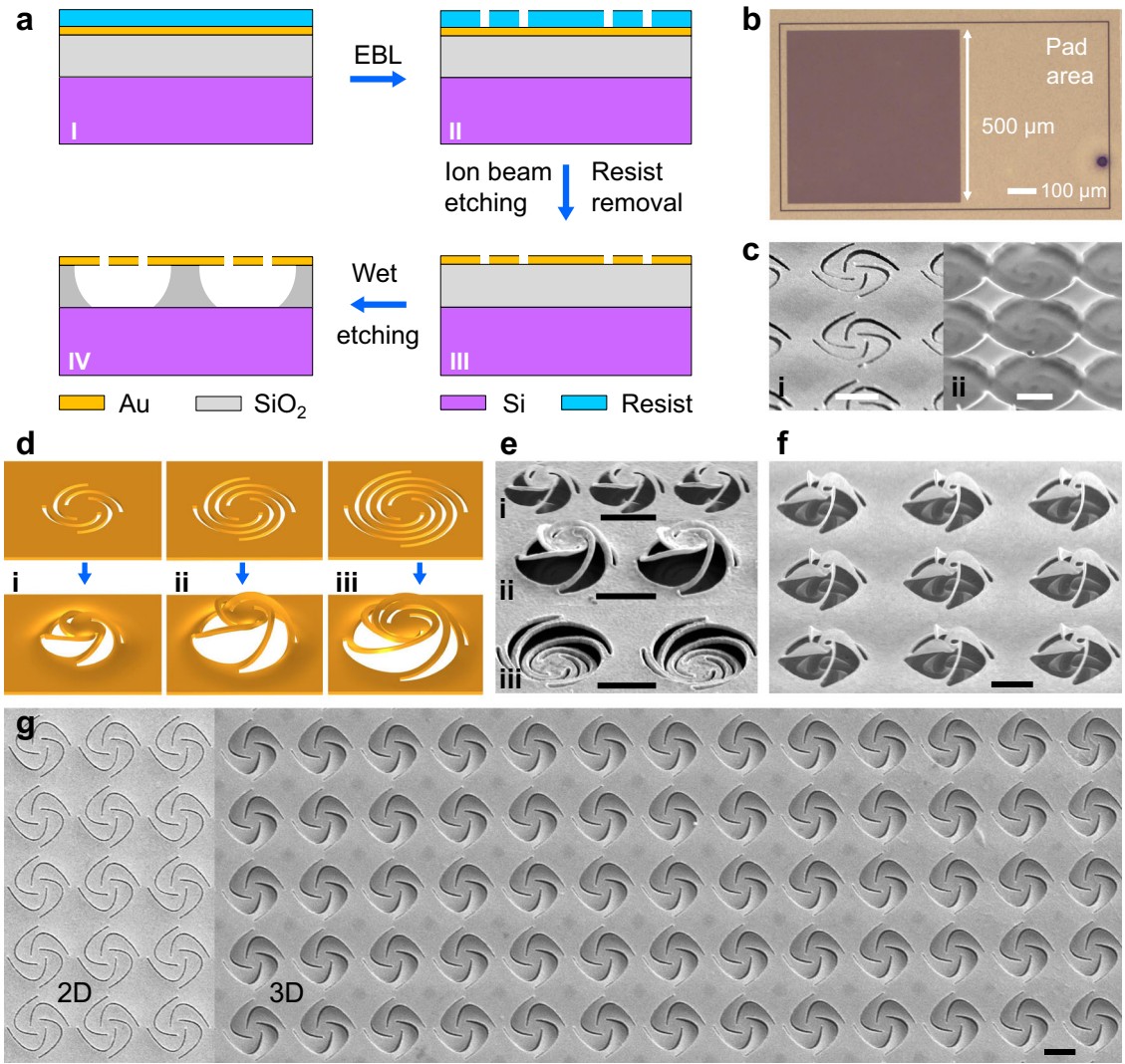

**Fig. 2 Sample fabrications. a** Flow chart of the fabrication process on an Au/SiO$_2$/Si chip. **b** Camera image of a pinwheel array with area of 500 × 500 μm$^2$ before wet etching. **c** SEM images of **i** pinwheels after wet etching and **ii** the below SiO$_2$ supporters after removing the top gold with FIB (Supplementary Fig. 2g). **d** Schematic of 2D and deformed 3D spirals in simulations. Image sizes: 2.5 × 2.5 μm$^2$. The 2D spirals consist of four arcs with angles of (i) 180°, (ii) 270°, and (iii) 360°, respectively, which are deformed into 3D by exerting upward stresses of 3, 3, and 1 GPa in simulations. **e, f** SEM images of three spirals and the pinwheels after wet etching and subsequent low-dose FIB irradiation. The images agree well with simulations except the type-iii spirals, which are stuck to the bottom substrate due to the capillary force and its weak stiffness (Supplementary Fig. 3). **g** Top-view SEM images of 2D and deformed 3D pinwheels under $V = 65$ V. Corresponding side-view images are shown in Fig. 1e–f. Structural parameters: $d = 300$ nm in **c**, **e**, **f** and $d = 500$ nm in **g**. Scale bars: 1 μm.

The second scheme of optical configuration is based on optical resonances excited in the nano-kirigami patterns, which is different from the simple cantilevers/membranes employed in conventional MEMS/NOEMS. In this case, both the unit cell and its periodicity are reduced to optical wavelength scales. As shown in Fig. 3e, by designing and measuring a spiral array with $w = 1.225$ μm and $p = 1.5$ μm (inset of Fig. 3f), a clear resonant dip in reflection spectrum is observed at wavelength $\lambda = 1842$ nm, where the gap plasmons are excited and form strong resonances confined by the curved slits (inset of Fig. 3e and Supplementary Fig. 6c, d). Due to the increased effective stiffness induced by the downscaling effects (Supplementary Fig. 1), such spirals with reduced size are difficult to achieve large deformations and the pull-in voltage is predicated at $V_p = 73$ V, as shown in Fig. 3d. As a result, only a tiny blue shift in spectra is observed when $V = 60$ V (Fig. 3e), which corresponds to a vertical structural deformation of ~70 nm. Nevertheless, in many realistic applications, it is the modulation contrast other than absolute values that is taken

into account. In this aspect, the modulation contrast of the fabricated spirals (defined as $\Delta R/R$) is found dramatically enhanced at the resonance wavelength, with a maximum modulation of 91% at $\lambda = 1860$ nm, as plotted in Fig. 3f. This strong modulation results from the high sensitivity of the plasmonic resonance modes confined within the curved slits (Supplementary Fig. 6c, d), which are drastically disturbed when the slits are deformed out of plane (inset of Fig. 3e). More importantly, such high-contrast modulation is reversible and the modulation reduces to zero instantly after turning off the electric voltage (Fig. 3f). Furthermore, the reversible reconfiguration of optical resonances is applicable to various nano-kirigami designs ($\lambda = 1500$ nm in Supplementary Fig. 8b) and the expected modulation frequency can be over 10 MHz (Supplementary Fig. 8c), exhibiting great potentials in electromechanical optical reconfigurations. It should be mentioned that both the modulation contrast (Supplementary Fig. 7c, d) and eigenfrequency (Supplementary Fig. 8c) can be altered by varying the amplitude

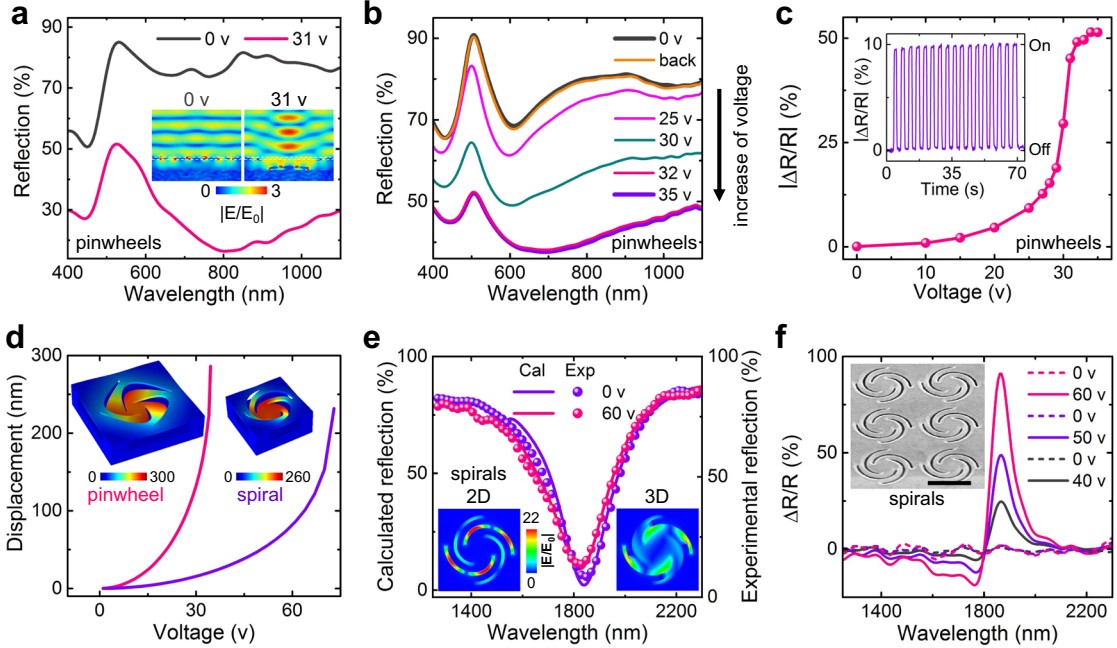

**Fig. 3 Electromechanically reconfigurable optical nano-kirigami. a** Calculated and **b** experimental reflection spectra in normal direction for a pinwheel array under different DC voltages as noted. Inset, calculated electric field distributions in the $xz$-plane ($y = 0$) under $V = 0$ and 31 V (with $\Delta d = 300$ nm and $\lambda = 750$ nm), respectively. Image size: $2.5 \times 2$ µm$^2$. The distorted wave shape at $V = 31$ V indicates the diffraction to other directions under deformations since $\lambda \ll w$ (see Supplementary Fig. 6a). In experiments, the reflection stops changing when $V > 32$ V and the spectrum increases back to the initial 0 V condition after turning off the voltage at 35 V. **c** Amplitude of modulation contrast (defined as $\left| \Delta R/R \right|$) versus applied voltage at $\lambda = 750$ nm. Inset, modulation contrast versus time when the voltage is turned on and off at 20 V and $\lambda = 550$ nm. **d** Calculated vertical displacement ($\Delta d$) versus applied voltage for a pinwheel and a type-i spiral, respectively, of which the pull-in voltages are identified at 35 and 73 V. Inset, simulated structures with corresponding maximum $\Delta d$ (units: nm). **e** Calculated (Cal) and experimental (Exp) reflection spectra of the type-i spirals in the inset of **f** under $V = 0$ and 60 V (with $\Delta d = 70$ nm), respectively. Inset, electric field distributions of the 2D and 3D spirals in $xy$-plane ($z = 0$) at $\lambda = 1842$ nm. Image size: $1.5 \times 1.5$ µm$^2$. **f** Modulation contrast versus wavelength when the DC voltage varies with a sequence $40 \to 0 \to 50 \to 0 \to 60 \to 0$ V (from bottom to top). Structural parameters: $w = 2$ µm, $p = 2.5$ µm, $d = 300$ nm for pinwheels; $w = 1.225$ µm, $p = 1.5$ µm, $d = 300$ nm for spirals. Scale bar: 1 µm.

of the applied voltage, offering a flexible way to engineer the electromechanical responses of the nano-kirigami structures.

**Reconfigurable submicron nano-kirigami.** Reducing the pixel size of on-chip reconfigurable optical devices is of great significance for high-resolution optical imaging, microscopy, fabrication, modulation, etc. The realization of submicron electromechanical elements is particularly challenged by the trade-off between the spatial miniaturization and the modulation depth, i.e., the high modulation depth requires large spatial displacements, while the submicron space limits the transformable range. Here, the electromechanical nano-kirigami with optical resonances provides an alternative solution with its flexible scaling feature. For example, a square array of cross wires with pitch size of 0.975 µm can be realized with our method, which exhibit well-defined double optical resonances in the near-infrared wavelength region (Fig. 4a). To reduce the operation voltage, the thickness of the SiO$_2$ supporters is chosen at $d = 200$ nm. Very interestingly, it is found that the reflection spectra are switchable between the states of $V = 20$ V and $V = -20$ V, as shown in Fig. 4a, with a maximum spectral shift of ~83 nm. The corresponding modulation contrast reaches 88% and 494% at wavelengths 953 and 1734 nm, respectively, as plotted in Fig. 4b. To the best of our knowledge, this is the smallest nano-kirigami structure with the highest modulation contrast that is achieved by electromechanical reconfiguration at near-infrared wavelengths.

It should be mentioned that different from the instant modulations in Fig. 3, the spectral modifications in Fig. 4a occur slowly and cannot return to the initial spectral position after

turning off the voltage, indicating a thermal-assisted plastic deformation process (see detailed analysis in Supplementary Fig. 9). Phenomenologically, when the isolated SiO$_2$ layer is very thin, thermal expansion induced by the underlying leakage current causes the red shift of the spectrum, while the electrostatic forces cause the blue shift of the spectrum (as observed in Fig. 3e). The competition between these two effects determines the final state of the structure. As P-doped silicon substrate is used in this work, the charge density at the space charge region in the case of $V > 0$ is much larger than that in the case of $V < 0$. This causes a stronger electrostatic force at $V = 20$ V, which results in the blue shift of the optical resonances compared with the case $V = -20$ V, as shown in Fig. 4a. Future in-depth investigation needs full analysis of the dynamics of the charges and the currents.

**Reconfigurable optical helicity.** Beyond the reconfiguration of light intensity, the electromechanical nano-kirigami is also promising for the modulation of other electromagnetic properties, such as polarization, phase, and helicity. For example, when the 2D planar patterns are deformed into 3D by the electrostatic forces, the mirror symmetry of the structures along the vertical direction is broken, which could result in enhanced helicity/chirality that universally exists in nature[45–47]. Importantly, it was found that optical helicity density fundamentally determines the circular dichroism (CD) in local interactions of light with chiral molecules or nanostructures[45], which is critical for the characterization of chiral molecules in important biochemistry and pharmaceutical industries[46,47]. Therefore, the realization of on-chip and active modulation on optical helicity density and the

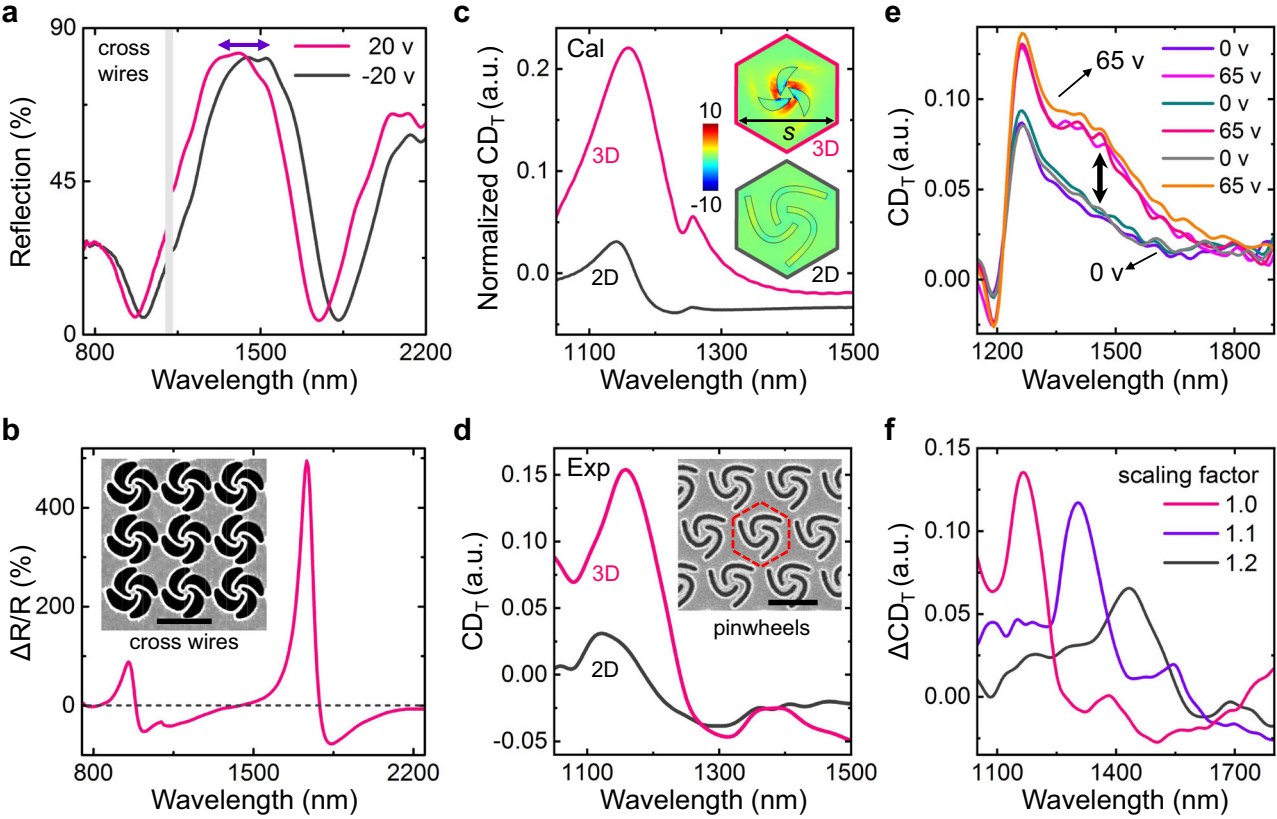

**Fig. 4 Submicron nano-kirigami and reconfigurable helicity. a** Measured reflection spectra of the cross wires in the inset of **b**, which are switchable between $V = 20$ and $-20$ V. **b** Measured modulation contrast in reflection spectrum for the cross wires in the inset. Structural parameters: $w = 0.796$ μm, $p = 0.975$ μm, $d = 200$ nm. **c**, **d** Simulated and measured CD spectra of initial 2D and deformed 3D three-arm pinwheels [defined as $CD_T$ in arbitrary units (a.u.), see "Methods"] at $V = 0$ and 60 V, respectively. To compare the changes induced by 3D deformations under the same starting condition, the simulated CD spectrum of the initial 2D pattern is normalized to the experimental spectrum of the same structure. Inset of **c**, calculated distribution of enhancement factor of optical helicity density for 2D (left, at $z = 0$ plane) and deformed 3D (right, at $z = -120$ nm plane) pinwheels in hexagonal unit cells under RCP incidence (Supplementary Fig. 10b, c). Inset of **d**, SEM image of the fabricated 2D pinwheels with scaling factor of 1.0. Structural parameters: $w = 0.880$ μm, pinwheel spacing $s = 1.15$ μm, $d = 300$ nm. **e** Measured CD spectra of three-arm pinwheels with $s = 1.265$ μm when the voltage is switched between 0 and 65 V (see Supplementary Fig. 10f). **f** Measured changes in CD spectra ($\Delta CD_T = CD_{T,3D} - CD_{T,2D}$) between 3D and 2D three-arm pinwheels under scaling factors of 1.0 (red), 1.1 (blue), and 1.2 (black). Scale bars: 1 μm.

associated CD is of great significance[48,49]. To this aim, an array of three-arm pinwheels are designed and the induced helicity enhancement factor is evaluated by $\Theta = \eta/\eta_0$, where $\eta$ and $\eta_0$ are the calculated optical helicity density with and without the nanostructure, respectively, under left-/right-handed circularly polarized (LCP/RCP) incidence (see "Methods" for details)[45]. As plotted in the inset of Fig. 4c, it is found that the $\Theta$ factor around the deformed 3D pinwheel is enhanced by more than one order of magnitude compared with that in the 2D pinwheel. Meanwhile, the associated CD spectra are calculated in Fig. 4c, where a dramatic enhancement is observed in the deformed 3D pinwheels. Such enhanced helicity and CD result from the broken mirror symmetry and the out-of-plane twisting, which induce handedness-dependent excitation of the electric quadrupole modes (see Supplementary Fig. 10a).

For experimental investigation, the designed 2D three-arm pinwheels are fabricated in a hexagonal lattice (inset of Fig. 4d), and the helicity-associated CD are characterized before and after the 3D deformations. As shown in Fig. 4d, the maximum CD of the 2D three-arm pinwheels is indeed prominently enhanced from 0.03 to 0.15 after the 3D transformations induced by the voltage, the trend of which is consistent with the simulation results in Fig. 4c. Moreover, it is found that this type of CD enhancement is dynamically tunable within the elastic

deformation range and highly scalable as the pinwheel width and periodicity are proportionally enlarged, as plotted in Fig. 4e, f, respectively. Such on-chip and electromechanical reconfiguration of optical helicity at optical wavelengths, while preliminary, opens a promising avenue for the exploration and application of interesting chiroptical phenomena.

## Discussion

In summary, an on-chip reconfigurable electromechanical nano-kirigami has been demonstrated at optical wavelengths. Large range of nano-kirigami-like 3D deformations and reversible modulation of optical responses have been clearly observed and readily engineered. Strong modulation of optical reflection and electromechanical modulation of optical helicity have been demonstrated in submicron nano-kirigami at near-infrared wavelengths. Importantly, such reconfigurable nano-electromechanical systems are compatible with the commercial nanofabrication technologies for further miniaturization and large-scale manufacturing. The proposed nano-kirigami principles and vertical actuation mechanism could be applied to a wide variety of material platforms for reconfigurable optical circuits and networks. For example, the top gold layer can be replaced by silicon (results not shown) and other conductive materials or

multilayer films, which could provide on-demand stiffness, Young's modulus, yield strength, etc.

It should be mentioned that the operation voltage of the electromechanical nano-kirigami can be reduced by decreasing the $SiO_2$ thickness or optimizing the nano-kirigami designs (Fig. 3d), which is understandable from $V = \sqrt{2k_{eff}\Delta d/\varepsilon A}(d - \Delta d)$ by applying $F_e = -F_r$. For example, for the same four-arm pinwheels, the pull-in voltages were found dropped from ~65 V (Fig. 2g) to 35 V (Fig. 3b), when $d$ decreased from 500 to 300 nm. Moreover, simulation results show that oblique incidence could affect the reflection and its modulation upon deformations (Supplementary Fig. 11), especially for the optically resonant nano-kirigami, which is natural since the excitation of plasmonic resonances is angle-dependent.

Based on the design principles, fabrication schemes, and optical evidences demonstrated in this work, further studies on the fast modulation dynamics (Supplementary Fig. 12) need full considerations of the strain field, velocity field, electromagnetic field, and charge density, which are interesting but are out of the scope of this article. Meanwhile, the reconfiguration experiments and optical measurements were all conducted in ambient atmosphere. Under such circumstance, the dynamic modulation frequency of the prototype nano-kirigami was measured up to 200 kHz (Supplementary Fig. 8d), although numerical simulations reveal that the fundamental resonant frequencies can be over 10 MHz for various nano-kirigami designs (see "Discussions" in Supplementary Fig. 8c).

Last but not the least, inheriting the merits of kirigami/origami, the operation bandwidth, modulation depth, and resonant frequency of the proposed reconfigurable nano-kirigmai can be readily engineered through the downscaling effects and synthetic designs. Such flexibility is very desirable for the realization of high-speed spatial light modulations with submicron pixels, which is promising since the pitch size and modulation speed of commercial DMD chips are limited to 5 μm and 40 kHz (ref. [50]), respectively. Therefore, the studies in this work could provide advanced methodologies and great flexibilities for future design and manufacture of on-chip optical manipulators, digital display devices, high-speed and high-resolution spatial light modulators, solid-state light detection and ranging, etc., which can find wide applications in areas of nano-photonics, optomechanics, MEMS, NOEMS, etc.

## Methods

**Numerical simulations**. Mechanical simulations of structural transformations were performed by using the finite element software COMSOL, and the electrostatic interactions between gold film and bottom silicon substrate was simulated to be the actuating force for the structural deformations. We implemented an implicit solver to bypass the numerical singularity around the device's pull-in voltage. As for frequency response, we conducted eigen-frequency analysis on the deformed structures under the applied voltages ranging from zero to near pull-in voltage. The reflection spectra of both initial 2D patterns and resulted 3D nano-geometries were simulated by using the finite-difference time-domain method. The $x$-polarized light was incident along the $z$-axis from the gold side and periodic boundary conditions were applied to the unit cell in the $x$–$y$ plane. Reflection spectra in normal direction were referenced to the blank Au/$SiO_2$/Si substrate. The optical helicity density $\eta$ was calculated in transmission mode with a simplified formula[45] $\eta = 2gRe(\nu\tilde{n}Im(H^* \cdot E))$, where $g = (16\pi\omega)^{-1}$, $\nu = n(r,\omega)/|n(r,\omega)|$, $\tilde{n} = \partial(\omega n(r,\omega))/\partial\omega.n(r,\omega)$ is the function of refractive index of the incident wave at frequency $\omega$, and $H$ and $E$ are the magnetic and electric fields, respectively. After $\eta$ and $\eta_0$ were calculated in a unit cell region with and without the nanostructure, respectively, under RCP incidence, the enhancement factor $\Theta = \eta/\eta_0$ was obtained and plotted in the inset of Fig. 4c.

**Sample fabrications**. For the 2D nanopatterning, commercial P-doped $SiO_2$/Si substrates (Lijingkeji Co., Ltd., 500 μm thick, P-doped) were firstly deposited with 5-nm-thick chromium and 60-nm-thick gold, spin-coated with 200-nm-thick poly (methyl methacrylate) resist, and baked at 180 °C for 1 min. Next, 2D nano-kirigami patterns were exposed by EBL and followed by ion beam etching of the gold. After the resist was removed, the sample was dipped into diluted hydrofluoric acid (40%, HF:$H_2O$ = 1:4) to etch off the beneath $SiO_2$, resulting in locally

suspended gold nanopatterns on the top. Such a fabrication procedure is standard and some of the EBL samples were provided by Tianjin H-Chip Technology with its own recipe. The fabricated samples were finally attached on the circuit board for electrical tests through either wire bonding or probe landing on the pad areas (Supplementary Fig. 5a, d). A dual-beam FIB/SEM system (FEI Helios 600i) was used to cut the sample at high dose (>600 pC μm$^{-2}$) or deform the suspended 2D patterns at low dose (10–40 pC μm$^{-2}$) for sample characterizations. The thickness of the $SiO_2$ layer was chosen at 500, 300, and 200 nm, respectively, by considering the availability of the chips and the proper structural deformation ranges. The areas of the nano-kirigami arrays were varied from $40 \times 40$ μm$^2$ to $500 \times 500$ μm$^2$ dependent on specific experimental purposes. Special care should be taken when using FIB/SEM systems to characterize the samples since they are invasive and may modify the optical and electromechanical properties of the nanostructures.

**Optical characterizations**. Optical measurements were conducted by using a homemade spectroscopy system. For reflection spectral measurement, visible white light from a tungsten halogen source (HL-2000, Ocean Optics) or a super-continuum light source (SC400-4, Fianium) was focused onto the sample by a near-infrared objective lens (×10, NA 0.25, Olympus). The reflected light in normal direction was collected by the same objective and delivered to a spectrometer (Ocean Optics, QE65000 for visible wavelengths and NIRQuest for near-infrared wavelengths). Due to the low sensitivity at the wavelength edges of the two spectrometers, a spectral gap appears around wavelength 1100 nm (Fig. 4a). For reconfiguration test, the DC voltage was supplied by a source meter (Keithley 2450). Generally, the pull-in voltages were observed at ~65 V when $d = 500$ nm and ~35 V when $d = 300$ nm, respectively, for the same four-arm pinwheels with width $w = 2$ μm. The pull-in state was also dependent on the specific 2D nano-kirigami patterns. For example, for the combined type-i spiral with $w = 1.225$ μm and $d = 300$ nm (Fig. 3f), the pull-in voltage exceeded 70 V. Therefore, cares need to be taken during the electrical tests when different nano-kirigami structures are investigated. To demonstrate the reconfigurable optical helicity, CD of the nano-kirigami structures under LCP and RCP incidence was measured by the same method as in ref. [24]. For simplicity and to avoid the influence from the silicon substrate (decreasing the transmission), the CD was defined in relative transmission by $CD_T = (T_{RCP} - T_{LCP})/(T_{RCP} + T_{LCP}) = (I_{RCP}/I_0 - I_{LCP}/I_0)/(I_{RCP}/I_0 + I_{LCP}/I_0) = (I_{RCP} - I_{LCP})/(I_{RCP} + I_{LCP})$, where $T$, $I$, and $I_0$ are the transmission, transmitted light intensity, and incident light intensity, respectively.

## Data availability

All the data supporting the findings of this study are available within the article, its Supplementary Information files, or from the corresponding author upon reasonable request.

## Code availability

Upon reasonable request, the authors will provide related code for the purpose to reproduce the results of this paper.

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

## Acknowledgements

The authors thank Yu Han, Xing Liu, Qianqin Jiang, and Junxiang Yan for helps in sample preparations and simulations. The authors thank Zhi-Yuan Li, Ling Lu, Xudong Zou, Yi Xu, Xiaoshan Zhu, Haifang Yang, Aizi Jin, Dongxiang Zhang, Qiulin Zhang, Laboratory of Optical Physics and Laboratory of Microfabrication at Institute of Physics (CAS), and Analysis & Testing Center at Beijing Institute of Technology for useful discussions and assistances in facility support. This work is supported by the National Natural Science Foundation of China (under Grant Nos. 61675227, 61975016, 61771402, 11704402, and 11674387), the National Key R&D Program of China (under Grant Nos. 2017YFA0303800, 2016YFA0200800, and 2016YFA0200400), the Science and Technology Project of Guangdong (2020B010190001), and Beijing Natural Science Foundation (Z190006).

## Author contributions

J.F.L. conceived the idea. J.F.L., C.C.T., and J.J.L. designed the initial fabrication scheme during 2018/01–2018/05. Z.G.L., C.C.T., R.H.P., L.C.Y., B.G.Q., and J.F.L. patterned some of samples with EBL and ion milling; S.S.C. and Z.G.L. conducted wet etching and wire bonding; S.S.C. and J.F.L. conducted optical measurements and reconfiguration tests, and analyzed the data; H.F.D., X.H.L., and N.X.F contributed to the mechanical modeling and reconfiguration discussions; C.Y.J., Z.G.L., and S.S.C. performed numerical simulations on optical properties; J.F.L., J.J.L., C.Z.G., Y.G.Y., X.D.Z., and N.X.F. supported and supervised the project. J.F.L. and S.S.C. wrote the manuscript. All authors participated in the project discussion and manuscript preparation.

## Competing interests

The authors declare no competing interests.
