## [Peer Review File · Nature Communications]

REVIEWER COMMENTS

Reviewer #1 (Remarks to the Author):

This manuscript presents the experimental investigation of the electromechanical actuation of optical nano-kirigami structures. The electrical tuning of the reflection spectrum and of the circular dichroism are demonstrated. While the work is innovative in terms of nanofabrication, I find the results in terms of optical functionality and especially their interpretation too thin to deserve publication in Nature Communications. The achieved modulation of the reflection spectrum (Fig. 3) is very limited, especially given the high voltages employed, and not competitive with other approaches. No attempt is made to interpret the observed features with electromagnetic simulations. The observed modulation of dichroism (Fig. 4) is potentially a more interesting feature of the proposed structure, however also here the experimental results are not compared to theory (the maps of optical helicity density do not allow gauging the expected performance and comparing with experiments). Overall, without a quantitative understanding of the observed effects, this work makes a limited contribution to the progress in the field. As a side note, I would encourage the authors to avoid using expressions like "CMOS compatible" while the materials (e.g. Au) employed in their work are clearly not compatible with CMOS technology

Reviewer #2 (Remarks to the Author):

Micro and nanoscale kirigami represent a forefront in adaptive optics and chiral photonics. Shanshan Chen and co-authors describe optical nano-electromechanical system based taking advantage of reconfigurable nano-kirigami. The salient features of their device design include utilization of electrostatic forces between the top gold nanostructure and bottom silicon substrate combined with CMOS compatibility and high modulation approaching 500%. Reversible kirigami deformations were observed that enabled variable helicity at the scale as small as 0.975 μm . The optical functionalities were observed at visible and near-infrared wavelengths.

The manuscript is significant and novel. It is recommended for publication with revisions.

While there is some identification of cyclability in Fig 3c and S6c, the data on cyclability need to be expanded. Also the cyclic performance under different voltages is likely to be different.

The optical properties of these devices (and all kirigami optics, in fact) will differ depending on the angle of light incidence. How does the angle of incidence change the reflectivity and its variability under bias, or instance?

It will be important to put the computational curves in Fig. 4c.

It seems to me Figure S9e should be in the main text.

Reviewer #3 (Remarks to the Author):

Recently, kirigami/origami provides new platforms for versatile advanced 3D microfabrication/nanofabrication. In the manuscript titled 'Electromechanically reconfigurable optical nano-kirigami', the authors report series of on-chip CMOS-compatible optical nano-kirigami, which can realize reversible optical helicity and high contrast optical modulation. The contribution of this work to the optical community can be summarized as follows: 1. The proposed nano-kirigami is on-chip, CMOS-compatible and integrable, which may yield device level applications. 2. The electromechanical based nano-kirigami can realize fast and accurate optical reconfiguration. I recommend it be published in Nature communication after addressing some of my comments.

1. To show the response time of reconfigurable nano-kirigami/origami, the authors should add some figures about vertical displacement versus time when applying external voltages, simulation results is fine.
2. Could the authors provide more physical explanations for the reflection change induced by shape transformation illustrated in figure 3(a). Since the incident light is not blocked totally, how about the transmission and absorption?

Dear Reviewers,

Thank you very much for reviewing our manuscript during the COVID-19 pandemic, which has caused a very difficult time to all of us. We are very pleased to see the supportive comments that “*the work is innovative in terms of nanofabrication*”, “*the manuscript is significant and novel*” and “*the electromechanical based nano-kirigami can realize fast and accurate optical reconfiguration*”. We also appreciate the reviewers’ valuable suggestions for further improvements. As a result, we have considered the remarks and made all necessary changes to fully address the comments. The followings are the responses to the comments.

Reviewer #1:

Comment 1: This manuscript presents the experimental investigation of the electromechanical actuation of optical nano-kirigami structures. The electrical tuning of the reflection spectrum and of the circular dichroism are demonstrated. While the work is innovative in terms of nanofabrication, I find the results in terms of optical functionality and especially their interpretation too thin to deserve publication in Nature Communications. The achieved modulation of the reflection spectrum (Fig. 3) is very limited, especially given the high voltages employed, and not competitive with other approaches. No attempt is made to interpret the observed features with electromagnetic simulations.

Response 1: Thank the reviewer very much for the positive comments on our nanofabrication results and sorry for the confusions in interpretations. In the revised manuscript, we have addressed the reviewer’s concerns from four aspects.

(I) First, the significance of the functionality in this work has been clarified. In the field of spatial light modulations, the reduction of pixel size (p) and the enhancement of modulation rate (f) are highly challenging, such as the widely commercialized liquid crystal spatial light modulators (LC-SLMs, with $f_{\max}=1$ kHz) and digital micromirror devices (DMDs, with $f_{\max}=40$ kHz and $p_{\min}=5$ μm) summarized recently in Ref. [*Light: Science & Applications* 8, 110 (2019)]. Moreover, optical elements with smaller pixels may lead to the manipulation of more complex polarization states, such as the recently reported full-Stokes polarization imaging [*Science* 365, 43 (2019)]. In this aspect, the optical nano-kirigami is promising with its small pixels [$p_{\min}=2.5$ μm (Fig. 3c), 1.5 μm (Fig. 3f) and 0.975 μm (Fig. 4b)] and fast modulation [$f_{\max}=200$ kHz (Supplementary Fig. S8d)]. Meanwhile, the nano-kirigami design, as a new means, holds

potentials for further development and improvement. For example, the modulation contrast can reach 494% by utilizing the sensitivity of the optical resonance at specific wavelength in Fig. 4b and the theoretical modulation frequency can be over 10 MHz (*Supplementary Fig. S8c*). To clarify this information, revisions have been made in [Lines 288-291, Page 13], as “*Such flexibility is very desirable for the realization of high-speed spatial light modulations (SLM) with submicron pixels, which is promising since the pitch size and modulation speed of commercial DMD chips are limited to 5 μm and 40 kHz⁵⁰, respectively.*”

(II) Second, with the prototypes demonstrated in this work, the operation voltage (V) can be further reduced through two ways. As illustrated in *the new Fig. 1c*, under the equilibrium state, the downward electrostatic force (F_e) and upward mechanical restoring force (F_r) reach a balance through $F_e = -\frac{1}{2}V^2 \frac{\epsilon A}{(d-\Delta d)^2} = -F_r = -k_{eff}\Delta d$, where A is the effective area, d and ϵ are the initial thickness and permittivity of the materials in between the plates, Δd is the absolute value of vertical displacement, and k_{eff} is the effective stiffness of the equivalent mass-spring system, respectively. Consequently, one can then get $V = \sqrt{2k_{eff}\Delta d/\epsilon A}(d - \Delta d)$, which means that V , d , Δd and k_{eff} are highly correlated. Therefore, **the first way** to decrease V is to reduce the thickness (d) of the SiO₂ supporters. For example, when d decreases, the operation voltage can be reduced from ~ 35 v (Fig. 3b) to 20 v (Fig. 4a) and the pull-in voltages drop from ~ 65 v (Fig. 2g) to 35 v (Fig. 3b). **The second way** is to reduce the effective stiffness (k_{eff}) of the displaced units through geometric optimization or material selection. For example, the pull-in voltage can be reduced from 73 to 35 v based on the geometric designs in Fig. 3d. To clarify this information, revisions have been made in [Lines 269-273, Page 12], as “*It should be mentioned that the operation voltage of the electromechanical nano-kirigami can be reduced by decreasing the SiO₂ thickness or optimizing the nano-kirigami designs (Fig. 3d), which is understandable from $V = \sqrt{2k_{eff}\Delta d/\epsilon A}(d - \Delta d)$ by applying $F_e = -F_r$. For example, for the same four-arm pinwheels, the pull-in voltages were found dropped from ~ 65 v (Fig. 2g) to 35 v (Fig.3b) when d decreased from 500 to 300 nm.*”

(III) The third feature of this work is that, different from conventional MEMS or NEMS, the nano-kirigami structures are optically resonant. This merit makes the optical responses more sensitive to the deformations than the non-resonant cases, and thereby remarkable modulation

contrast can be easily obtained. For example, a 70-nm deformation resulted in a modulation contrast of 91% near the optical resonances (Fig. 3f). In comparison, a 300-nm deformation only resulted in a modulation contrast of 50% in the non-resonant case (Fig. 3c). This feature could be very useful for the generation of versatile electrical, mechanical and optical resonances, as well as their inter-couplings. To clarify this information, revisions have been made in [Lines 184-186, Page 8], as “*This strong modulation results from the high sensitivity of the plasmonic resonance modes confined within the curved slits (Supplementary Figs. S6c-6d), which are drastically disturbed when the slits are deformed out of plane (inset of Fig. 3e).*”

(IV) As suggested by the reviewer, corresponding electromagnetic simulations have been added to interpret the observed results, such as in Fig. 3a, inset of Fig. 3e, Fig. 4c, and Supplementary Figs. S6 and Fig. S10a (highlighted in Supplementary Information), such as:

Descriptions of Fig. 3a and Supplementary Fig. S6a: “*To test this scheme, a pinwheel array with periodicity of 2.5 μm is simulated under normal incidence at visible wavelengths. As plotted in Fig. 3a, the reflection spectrum drops significantly when the 2D pinwheels are deformed into 3D with a height of 300 nm under a voltage of 31 v, which is mainly caused by the diffraction to other directions, as shown by the inset of Fig. 3a and Supplementary Fig. S6a.*” in [Lines 147-152, Page 7].

Fig. 3 (a) Calculated reflection spectra in normal direction for a pinwheel array under different DC voltages as noted. **Inset**, calculated electric field distributions in the xz -plane ($y=0$) under $V=0$ and 31 v (with $\Delta d=300$ nm and $\lambda=750$ nm), respectively. Image size: $2.5 \times 2 \mu\text{m}^2$. The distorted wave shape at $V=31$ v indicates the diffraction to other directions under deformations since $\lambda \ll w$ (see Supplementary Fig. S6a). **(e)** **Inset**, electric field distributions of the 2D and 3D spirals in xy -plane ($z=0$) at $\lambda=1842$ nm. Image size: $1.5 \times 1.5 \mu\text{m}^2$.

Inset of **Fig. 3e** and *Supplementary Figs. S6c-6d*: “a clear resonant dip in reflection spectrum is observed at wavelength $\lambda=1842$ nm, where the gap plasmons are excited and form strong resonances confined by the curved slits (inset of Fig. 3e and Supplementary Figs. S6c-6d).” in [Lines 173-175, Page 8].

Fig. 4c and *Supplementary Fig. S10a*: “Meanwhile, the associated CD spectra are calculated in Fig. 4c, where a dramatic enhancement is observed in the deformed 3D pinwheels. Such enhanced helicity and CD result from the broken mirror symmetry and the out-of-plane twisting, ...handedness-dependent excitation of the electric quadrupole modes (see Supplementary Fig. S10a).” in [Lines 240-244, Page 11, see details in below **Response 2**].

Comment 2: The observed modulation of dichroism (Fig. 4) is potentially a more interesting feature of the proposed structure, however also here the experimental results are not compared to theory (the maps of optical helicity density do not allow gauging the expected performance and comparing with experiments).

Response 2: Thank the reviewer for the positive comments on the modulation of optical circular dichroism. To address the reviewer’s concern, numerical simulations have been added in the revised manuscript as the new Fig. 4c. The spectral features of both theoretical (Fig. 4c) and experimental (Fig. 4d) results are consistent, while the amplitudes differ due to the imperfections in fabrications and measurements. The physical explanation of the observed CD enhancement, i.e. the handedness-dependent excitation of the electric quadrupole modes induced by the out-of-plane twisting, has been uncovered by the electromagnetic multipolar analysis.

Fig. 4 (c,d) Simulated and measured CD spectra (defined as CD_T , see Methods) of initial 2D and deformed 3D three-arm pinwheels at $V=0$ and 60 v, respectively. To compare the changes induced by 3D deformations under the same starting condition, the simulated CD spectrum of the initial 2D pattern is normalized to the experimental spectrum of the same structure.

Fig. S10. (a) Calculated scattering power from various multipolar moments induced in the 2D and 3D three-arm pinwheels in Fig. 4c under RCP and LCP incidence... The red spectral peaks of the 3D pinwheels indicate the handedness-dependent excitation of EQ modes, which induces the CD enhancement.

To clarify this information, revisions have been made in [Lines 240-244, Page 11], as “Meanwhile, the associated CD spectra are calculated in Fig. 4c, where a dramatic enhancement is observed in the deformed 3D pinwheels. Such enhanced helicity and CD result from the broken mirror symmetry and the out-of-plane twisting, which induce handedness-dependent excitation of the electric quadrupole modes (see Supplementary Fig. S10a).”

Comment 3: Overall, without a quantitative understanding of the observed effects, this work makes a limited contribution to the progress in the field.

Response 3: Thank the reviewer for the important comments which guide us towards the better understanding and interpretation of the observed effects. As mentioned in above **Response 1**, we have addressed this comment from three aspects.

First, a new sub-figure about the physical picture of the reconfigurable optical nano-kirigami has been added in the new Fig. 1c for better understanding. In particular, we show that the effective stiffness can be flexibly engineered by the topology designs of the nano-kirigami patterns (see Supplementary Fig. S1). This provides a new freedom to engineer the optical and mechanical properties of the electromechanical nanostructures.

Fig. 1 (c) A simplified electromechanical model of the reconfigurable nano-kirigami, in which the displacement of the suspended nanostructure is controlled by the downward electrostatic force (F_e) and upward mechanical restoring force (F_r).

Second, considering the irregular geometries of the nano-kirigami patterns and complexity of the electromechanical system, numerical simulations and calculations have been conducted for the quantitative understanding of the observed effects. For example, Fig. 3a, inset of Fig. 3e, and Supplementary Figs. S6c-6d have been added to illustrate the deformations induced changes in electromagnetic responses (see above **Response 1-IV**). Meanwhile, the physical origins of the modulation in optical circular dichroism have been identified as the handedness-dependent excitation of the electric quadrupole modes (see Fig. 4c, Supplementary Fig. S10a and details in above **Response 2**). Moreover, theoretical calculations and discussions on dynamic modulation have been added as Supplementary Fig. S12 (see details in below **Response 2 of Reviewer #3**).

Third, the prototype designs and nanofabrication explorations provide new methodologies towards the miniaturization and fast modulation of reconfigurable optical elements, which hold potentials in developing high-speed and miniaturized spatial light modulations that are challenging with LC-SLMs and DMDs. This point has been clarified in above **Response 1-I**.

From these points of view, our work could be useful for the advancement in structural designs, physics exploration and device applications in micro-/nanophotonics, MEMS, NOEMS, etc., especially in the development of high-speed and miniaturized modulations of spatial light.

Comment 4: As a side note, I would encourage the authors to avoid using expressions like “CMOS compatible” while the materials (e.g. Au) employed in their work are clearly not compatible with CMOS technology.

Response 4: Sorry for the confusions. In the original draft, we meant that the fabrication method itself is “CMOS compatible” if the Au film can be replaced by other conductive and reflective materials compatible with CMOS technology. Actually, this fabrication method has been testified successfully by us on silicon-on-insulator (SOI) substrate (results not shown). To clarify this information, the description has been revised in [Lines 105-106, Page 5], as “*which is compatible with the complementary metal-oxide-semiconductor (CMOS) technique when the Au layer is replaced by proper conductive nanofilms (such as Si).*” And to avoid confusions, the words “CMOS compatible” in other places have been avoided.

Reviewer #2

Comment 1: Micro and nanoscale kirigami represent a forefront in adaptive optics and chiral photonics. Shanshan Chen and co-authors describe optical nano-electromechanical system based taking advantage of reconfigurable nano-kirigami. The salient features of their device design include utilization of electrostatic forces between the top gold nanostructure and bottom silicon substrate combined with CMOS compatibility and high modulation approaching 500%. Reversible kirigami deformations were observed that enabled variable helicity at the scale as small as 0.975 μm . The optical functionalities were observed at visible and near-infrared wavelengths.

The manuscript is significant and novel. It is recommended for publication with revisions.

Response: Thank the reviewer very much for the positive comments on our manuscript.

Comment 2: While there is some identification of cyclability in Fig 3c and S6c, the data on cyclability need to be expanded. Also the cyclic performance under different voltages is likely to be different.

Response: Thank the reviewer for this important comment. As suggested, the data on cyclability has been expanded in *Supplementary Fig. S7c-7e* of the revised manuscript as below.

Fig. S7. (c,d) Modulation contrast at wavelength $\lambda = 550$ nm versus time when the voltage is manually turned “on” and “off” for: (c) 5 cycles under $V = 10$ v; (d) 20 cycles under $V = 20$ v. (e) Modulation contrast under different frequencies as noted when the actuation voltage is controlled by a function signal generator under $V = 10$ v and $\lambda = 550$ nm.

Meanwhile, as the reviewer mentioned, the cyclic performance under different voltages is different from two aspects. First, the modulation contrast, determined by the vertical displacement (Δd), is dependent on the applied voltage since $F_e = -\frac{1}{2}V^2 \frac{\epsilon A}{(d-\Delta d)^2}$, i.e. a higher

voltage V results in a stronger F_e , and consequently a larger Δd with stronger modulation. This effect has been verified by the experimental results in *Supplementary Figs. S7c-7d*, where the modulation contrast under $V=20$ v is more than two times of that under $V=10$ v.

Second, the mechanical resonant frequency f_m is also dependent on the applied voltage since $f_m = \frac{1}{2\pi} \sqrt{k_{eff}/m_{eff}}$ and $k_{eff} \propto \frac{V^2}{\Delta d(d-\Delta d)^2}$ [since $V = \sqrt{2k_{eff}\Delta d/\epsilon A(d-\Delta d)}$], where k_{eff} and m_{eff} are effective stiffness and mass of the equivalent mass-spring system, respectively. Such effects have been verified by numerical simulations in *Supplementary Fig. S8c.*, where f_m decreases from 12 to 9 MHz for the spirals when V increases from 10 to 64 v.

To address this comment, a description has been added in [Lines 191-195, Page 9], as “*It should be mentioned that both the modulation contrast (Supplementary Figs. S7c-S7d) and eigenfrequency (Supplementary Fig. S8c) can be altered by varying the amplitude of the applied voltage, offering a flexible way to engineer the electromechanical responses of the nano-kirigami structures.*”

Comment 3: The optical properties of these devices (and all kirigami optics, in fact) will differ depending on the angle of light incidence. How does the angle of incidence change the reflectivity and its variability under bias, or instance?

Response 3: Thank the reviewer for this important comment. It is true that the optical properties of the nano-kirigami structures are dependent on the angle of incident light. Specifically, for the non-resonant optical modulations, the increase of incident angle results in the blue shift of the modulation band, with modulation peak values slightly changed. In comparison, for the resonant optical nano-kirigami, the resonant narrow-band modulation is red shifted with the increase of incident angle, while the peak value changes dramatically under large angles.

To address this comment, a description has been added in [Lines 273-276, Page 12], as “*Moreover, simulation results show that oblique incidence could affect the reflection and its modulation upon deformations (Supplementary Fig. S11), especially for the optically resonant nano-kirigami, which is natural since the excitation of plasmonic resonances is angle-dependent.*” Meanwhile, a new *Supplementary Fig. S11* (see below) has been added in the *Supplementary Information*.

Fig. S11. Simulation results under oblique incidence. (a, c) Calculated reflection spectra of 2D and deformed 3D **(a)** four-arm pinwheels (as in Fig. 3a) and **(c)** spirals (as in Fig. 3e) under oblique incident angle of 15°, 30°, 45°, respectively. **(b)** Amplitude of modulation contrast (defined as $|\Delta R/R|$) versus wavelength for the pinwheels under different incident angle as noted.... **(d)** Modulation contrast versus wavelength for the spirals under different incident angle as noted...

Comment 4: It will be important to put the computational curves in Fig. 4c.

Response 4: As suggested by the reviewer, the new computational curves have been added in Fig. 4c and a description has been added in the revised manuscript, as “*Meanwhile, the associated CD spectra are calculated in Fig. 4c, where a dramatic enhancement is observed in the deformed 3D pinwheels.*” in [Lines 240-241, Page 11] (also see **Response 2** of **Reviewer #1**).

Comment 5: It seems to me Figure S9e should be in the main text.

Response 5: Thanks for the suggestion. The *Supplementary* Fig. S9e has been adjusted to be added in the main text as **the new Fig. 4e**. The caption of Fig. 4e has been added. Note that “ $p=0.996 \mu\text{m}$ ” in the caption of original Fig. 4c was referred to the vertical lattice periodicity, which has been corrected as the inter-spacing of the submicron pinwheels for clearness, i.e. “ $s=1.15 \mu\text{m}$ ” ($p = \frac{\sqrt{3}}{2} s$ in the hexagonal lattice), as the arrow noted in the inset of Fig. 4c.

Reviewer #3

Comment 1: Recently, kirigami/origami provides new platforms for versatile advanced 3D microfabrication/nanofabrication. In the manuscript titled ‘Electromechanically reconfigurable optical nano-kirigami’, the authors report series of on-chip CMOS-compatible optical nano-kirigami, which can realize reversible optical helicity and high contrast optical modulation. The contribution of this work to the optical community can be summarized as follows: 1. The proposed nano-kirigami is on-chip, CMOS-compatible and integrable, which may yield device level applications. 2. The electromechanical based nano-kirigami can realize fast and accurate optical reconfiguration. I recommend it be published in Nature communication after addressing some of my comments.

Response 1: Thank the reviewer very much for the positive comments on our manuscript.

Comment 2: To show the response time of reconfigurable nano-kirigami/origami, the authors should add some figures about vertical displacement versus time when applying external voltages, simulation results is fine.

Response 2: Thank the reviewer for this important comment. Actually, the direct transient numerical analysis of high-frequency (above MHz) electrically actuated device is computationally challenging. This is because the strain field in the solids, velocity field of the air and electromagnetic fields in all metal-dielectric-fluid composites need to be solved simultaneously to simulate the coupled dynamics accurately. In particular, the time complexity increases dramatically when the resonance frequency of structure exceeds the inverse of relaxation time of structure by orders of magnitude.

To bypass these challenges and address the reviewer’s comment, we first computed the small-signal frequency response of the whole system within a wide range (*Supplementary Fig. S12a*), where the driving voltage was set with a bias of $V_0=10$ v and an undulating amplitude of $\Delta V=0.1$ v. After that, we performed frequency-to-time transformation and derived the time response of the electromechanical nano-kirigami. More specifically, we demonstrated the feasibility of this approach by applying a Standard Linear Solid Model to characterize the viscoelastic behavior of materials under cyclic loading (as in the inset of *Supplementary Fig. S12a*), although this special process could increase the calculated response frequency. Finally, the expected time response of the system, i.e. the vertical perturbation displacement versus time, is obtained in *Supplementary Fig. S12b*. We argue that this periodic response of the structural composite is analogous to the steady-state oscillation of the real system when the dissipation

of energy is compensated by the supply from the driving electrostatic force. The transition from initial system at rest to this steady state vibration requires a fully-coupled model from scratch with extremely highly-resolved time steps and carefully-calibrated material dynamic properties, which will be the focus of our future work but is out of the scope of this manuscript.

To address this comment, a new Fig. S12 and corresponding discussions have been added in *Supplementary Information* (see below). Corresponding description has been added in the main text [Lines 278-280, Page 12], as: “*further studies on the fast modulation dynamics (Supplementary Fig. S12) need full considerations of the strain field, velocity field, electromagnetic field and charge density, which are interesting but are out of the scope of this article.*”

Fig. S12. Calculations on dynamic modulation properties. (a) Small-signal frequency response of the nano-kirigami structure ($p=2.5 \mu\text{m}$) at $V_0=10 \text{ v}$ biased position and an undulating amplitude of $\Delta V=0.1 \text{ v}$... The sharp peak corresponds to the resonant frequency of the damped system. **Inset**, schematic illustration of the Standard Linear Solid Model in calculations... (b) Vertical perturbation displacement versus time by applying frequency-to-time transformation of the response curve in (a) (see below discussions).

Discussions on time-domain responses. Direct transient numerical analysis of high-frequency (above MHz) electro-actuated device are computationally challenging. This is because the strain field in the solids, velocity field of the air and electromagnetic fields in all metal-dielectric-fluid composites need to be solved simultaneously to simulate the coupled dynamics accurately [3-4]the focus of our future work but is out of the scope of this work.

Comment 3: Could the authors provide more physical explanations for the reflection change induced by shape transformation illustrated in figure 3(a). Since the incident light is not blocked totally, how about the transmission and absorption?

Response 3: The reviewer is right. Our simulations show that in the case of non-resonant optical modulation, the decrease in reflection along the normal direction at visible wavelengths is mainly caused by the increased high-order diffraction to other directions due to $\lambda \ll w$, while the transmission is slightly increased and the change in absorption is negligible. In comparison, in the case of resonant optical modulation, the change in reflection at near-infrared wavelengths is caused by both the decrease in absorption and the increase in transmission where $\lambda > w$.

To address this comment and provide physical explanations, the schematic illustration in the original Fig. 3a is replaced by quantitative simulations in the revised manuscript, such as the simulation spectra and field distributions in the new Fig. 3a, as well as the simulated optical spectra in the new *Supplementary Figs. S6a-6b* (see below). Corresponding descriptions have been added in [Lines 147-152, Page 7], as “*To test this scheme, a pinwheel array with periodicity of $2.5 \mu\text{m}$ is simulated under normal incidence at visible wavelengths. As plotted in Fig. 3a, the reflection spectrum drops significantly when the 2D pinwheels are deformed into 3D with a height of 300 nm under a voltage of 31 v, which is mainly caused by the diffraction to other directions, as shown by the inset of Fig. 3a and Supplementary Fig. S6a.*”

Fig. 3 (a) Calculated reflection spectra in normal direction for a pinwheel array under different DC voltages as noted. **Inset**, calculated electric field distributions in the xz -plane ($y=0$) under $V=0$ and 31 v (with $\Delta d=300$ nm and $\lambda=750$ nm), respectively. Image size: $2.5 \times 2 \mu\text{m}^2$. The distorted wave shape at $V=31$ v indicates the diffraction to other directions under deformations since $\lambda \ll w$ (see Supplementary Fig. S6a).

Fig. S6. (a) Calculated transmission, absorption and high-order diffraction spectra of 2D and deformed 3D pinwheels (as in Fig. 3a, $\lambda \ll w$) under normal incidence. It can be seen that after the 3D deformations, the high-order diffraction to non-normal directions (red curves) is dramatically enhanced and the near-infrared transmission is slightly increased, while the change in absorption is negligible. **(b)** Calculated transmission and absorption spectra of 2D and deformed 3D spirals (as in Fig. 3e, $\lambda > w$) under normal incidence. The 3D deformations induce enhanced transmission and reduced absorption in the near-infrared wavelengths.

Other Changes:

- (1) The orders of the figures have been updated.
- (2) The Acknowledgments section has been updated.
- (3) Some typos have been corrected and highlighted.
- (4) Information about the measurement systems has been added and highlighted in the captions of *Supplementary Fig. S7e* and *Fig. S8d*.

Please let me know if you need any further information.

Thank you very much.

Best regards,

Jiafang Li, Junjie Li and Nicholas X. Fang

Corresponding Emails:

jiafangli@aphy.iphy.ac.cn; jjli@aphy.iphy.ac.cn; nicfang@mit.edu

REVIEWERS' COMMENTS

Reviewer #2 (Remarks to the Author):

Looking good!

Reviewer #3 (Remarks to the Author):

All my concerns have been addressed in the revised manuscript. Now I would like to recommend the paper for publication.

Dear Reviewers,

Thank you very much for re-reviewing our manuscript (ID: NCOMMS-20-15048A). We are very pleased to see that all the reviewers recommend the paper for publication in *Nature Communications*. The followings are the responses to the comments.

Reviewer #2 (Remarks to the Author): Looking good !

Response: Thank the reviewer very much for the positive comment on our manuscript.

Reviewer #3 (Remarks to the Author): All my concerns have been addressed in the revised manuscript. Now I would like to recommend the paper for publication.

Response: Thank the reviewer very much for the recommendation. We are pleased to see that the revised manuscript has addressed all the reviewer's concerns.

Other Changes:

(1) There is a typo in the name of the 8th author in the manuscript, which has been corrected from "Lecheng Yang" to "**Lechen** Yang" in the revised manuscript. The author information in the submission system is correct and without change.

(2) The format of Main text and the Supplementary Information has been updated according to the journal style and the requirements in the Author-checklist file, which is highlighted in red color in the text.

(3) Few typos have been corrected and highlighted in the revised manuscript.

Please let me know if you need any further information.

Thank you very much.

Best regards,

Jiafang Li, Junjie Li and Nicholas X. Fang

Corresponding Emails:

jiafangli@aphy.iphy.ac.cn; jjli@aphy.iphy.ac.cn; nicfang@mit.edu